# Transforaminal pulsed radiofrequency and epidural steroid injection on chronic lumbar radiculopathy: A prospective observational study from a tertiary care hospital in Vietnam

**Viet-Thang Le**[1,2]☯**, Phuoc Trong Do**[1]☯**, Vu Duc Nguyen**[1‡]**, Luan Trung Nguyen Dao**[1‡]*

**1** Department of Neurosurgery, University Medical Center, Ho Chi Minh City, Vietnam, **2** Faculty of Medicine, University of Medicine and Pharmacy, Ho Chi Minh City, Vietnam

☯ These authors contributed equally to this work.
‡ These authors also contributed equally to this work.
* luan.dnt@umc.edu.vn

**Data Availability Statement:** All relevant data are within the paper and its Supporting Information files.

## Abstract

### Background

Lumbosacral radicular syndrome (LRS) is probably the most frequent neuropathic pain syndrome, exaggerating medical and economy burden on developing countries, such as Vietnam. As a result, the urgence to find an approach which is both affordable and effective always puts great demand on medical researchers.

### Objectives

Evaluate the effectiveness of transforaminal pulsed radiofrequency (PRF) stimulation on the dorsal root ganglion (DRG) and epidural steroid injection (ESI) in management of chronic lumbosacral radiculopathy.

### Methods

Seventy-six patients with chronic radicular pain were performed transforaminal PRF + ESI by neurosurgeons. Demographic characteristics and surgical outcomes were recorded on admission, pre-procedural and post-procedural for 1-month, 3-month, 6-month and 12-month follow-up. Primary outcome was measured by using Visual Analogue Scale (VAS), Oswestry disability index (ODI) and Straight Leg Raising Test (SLRT). Secondary outcome was subjectively collected based on short assessment of patients' satisfaction (SAPS).

### Results

Patients who received transforaminal PRF and ESI showed significant improvements on all three evaluation tools (VAS, ODI, SLRT), compared to that before treatment (p<0.001). Pain relief was achievable and long-lasting, which met patients' expectation. No significant complications were observed for 12 months follow-up.

**Funding:** The author(s) received no specific funding for this work.

**Competing interests:** The authors have declared that no competing interests exist.

**Abbreviations:** PRF, Pulsed radiofrequency ablation; DRG, dorsal root ganglion; VAS, Visual Analogue Scale; ODI, Oswestry Disability Index; ESI, epidural steroid injection; MRI, magnetic resonance imaging.

## Conclusion

Transforaminal PRF combined with ESI in management of lumbosacral radiculopathy should be a good method of choice for its effectiveness and safety in management of pain.

## Introduction

Lumbosacral radicular syndrome (LRS) is probably the most frequent neuropathic pain syndrome and 30% of patients suffer chronic pain which lasts more than 3 months [1,2]. Chronic lumbar radiculopathy is a complex condition that requires multimodality and multidisciplinary attention. In recent years, pulsed radiofrequency (PRF) has emerged and appeared to be an effective noninvasive technique that relies on the intermittent administration of high frequency current. It offers pain relief without causing tissue damage [3].

On the other hand, epidural steroid injection (ESI) has become a common procedure in interventional pain practices [4]. ESI offers the strong effect of blocking nociceptive C-fiber conduction and effectively reducing nerve root inflammation. In short term, ESI is widely accepted and conventionally used. In long-term, however, effects can be unsatisfactory.

Based on body of recent literature, we remarked that transforaminal PRF on the dorsal root ganglion (DRG) combining with ESI in management of chronic lumbosacral radiculopathy achieved positive results with long-lasting effects. Yang et al [5] reported that after a median follow-up of 9.5 months, the VAS decreased from 6.5 ± 1.0 to 2.4 ± 1.9 at the last follow-up. Karakos et al showed that VAS in the transforaminal ESI + PRF group were significantly lower than the transforaminal ESI group at all follow-up periods (p<0.001).

In Vietnam, pain management remains a new and uncharted medical field. Plus, the state of economic development in general prevents patients and even medical workers accessing advancing healthcare. The burden of chronic pain, particularly chronic lumbar radiculopathy, takes shape and malignantly grows. As a result, this study was conducted and aimed to evaluate the efficacy and safety of transforaminal PRF and ESI, as well as its potential for the management of chronic lumbosacral radicular pain.

## Materials and methods

### Study design, setting and participant

A prospective observational study was conducted to evaluate the efficacy of transforaminal PRF and ESI for management of chronic lumbar radicular pain. Patients had physical and neurological examination, as well as laboratory tests. All operation was performed at the operating theatre in University Medical Center, Ho Chi Minh City, Vietnam. From November 2019 to November 2021, seventy-six patients were enrolled based on the inclusion and exclusion criteria. Inclusion criteria were as follows:

- Age > 18 years

- Clinical manifestation:

- Pain level on VAS ≥ 5

- Low back pain for more than 3 months and unilateral lower limb with radicular pain.

- No history of spinal surgery.

- Posterolateral lumbosacral disc herniation on magnetic resonance imaging (MRI) and corresponding clinical symptoms and signs.

- All patients presented who refused open surgery and were ineffective conservative treatment after three months, including physical therapy, manipulation, and non-morphine treatment (Gabapentinoids, Duloxetine, Tramadol, Nonsteroidal anti-inflammatory drugs).

   Exclusion criteria were as follows:

- Spinal tumors, tuberculosis, spondylolisthesis (grade $\geq$ II), and fracture spine.

- Symptoms of severe nerve damage include motor paralysis, muscle atrophy, cauda equina syndrome, uncontrolled diabetes mellitus, uncontrolled hypertension, cardiac diseases, malignancy, and bleeding diathesis.

- Multi-segmental lumbar disc herniation, spinal deformity.

- Infection.

- Coagulation disorder.

- Prior radiofrequency (RF) treatment for low back pain.

- Prior epidural injection in the past 3 months, such as nerve root injection and caudal injection.

- Pregnancy.

- Allergic to drugs used in the procedure.

## Study procedure

The patients were in a prone position for C-arm fluoroscopy (GE Healthcare, USA) and under local anesthesia. To determine the point for injection: 3-5cm from the midline, depending on the side with related symptoms, check the corresponding spinal segment below the C-arm. Align the C-arm head in the oblique direction until the superior articular surface of the lower vertebrae positioning in the middle of the disc space. The classic image described as "Scotty Dog" will be displayed. Perform local anesthesia with 1% lidocaine and gently insert the 20-gauge tip cannula (RF cannula needle, length 100 mm or 150 mm with a 10 mm active tip, Cosman Medical, Massachusetts, USA) until it passes the outer edge of the superior joint. Stepwise procedure was conducted according to the practice guidelines of the International Spine Intervention Society [6]:

- Adjust the lamp head in the front and back direction, and determine whether the needle tip was on the inter-peduncle space (Fig 1).

- Align the lamp head in the lateral direction; locate the needle tip at the lower back of vertebrae foramen. Then, the depth of the needle tip is checked by the lateral view of the C-arm (Fig 2).

- Gently withdraw the syringe to check for blood or cerebrospinal fluid. After entering the epidural area, 1 ml of contrast agent (Ultravist, Schering AG. Berlin, Germany) was injected into the anterior epidural region. Relating nerve root was visualized (Fig 3). Any sign of arterial/venous intravasation of the contrast agent was controlled through simultaneous imaging. Exemplary case showed on S1 and S2 Figs.

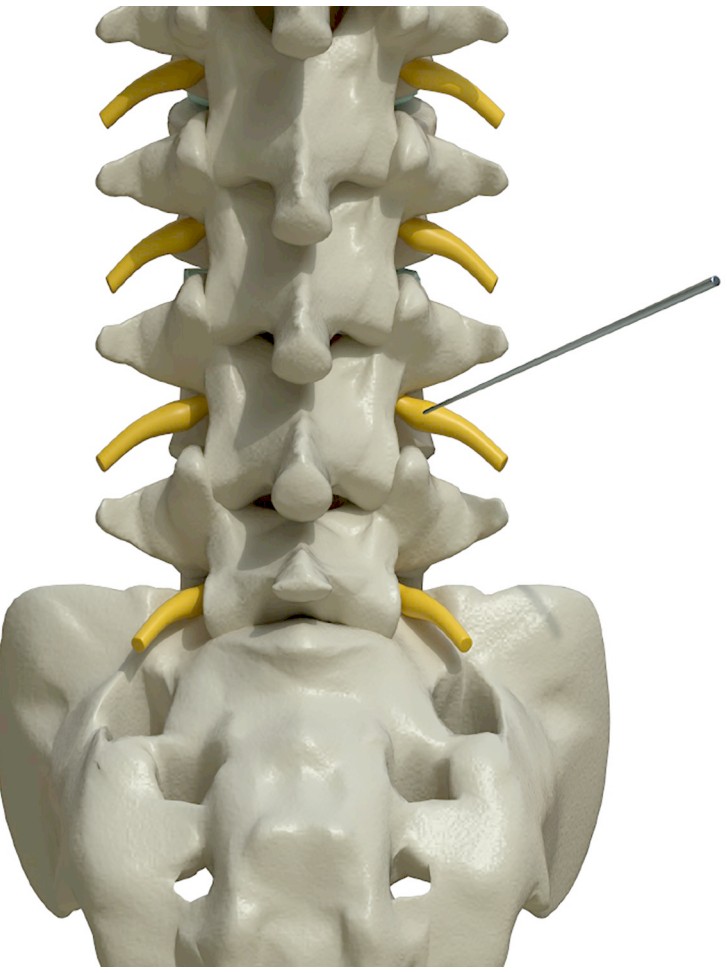

**Fig 1. 3D illustration of transforaminal PRF+ESI with anteroposterior view.** The needle tip is on the inter-peduncle space and adjacent to dorsal root ganglion of left-side L4 nerve root. Copyright kieutrinhnguyen. Published with permission.

- The catheter needle (active tip electrode) was inserted and a sensory stimulation test was carried out using an RF generator (Cosman G4, Cosman Medical, Massachusetts, USA). The catheter needle was then advanced toward the dorsal root ganglion until the patient reported a tingling sensation and/or dysesthesia at less than 0.3 Volts. PRF treatment was administered at 5 Hz and 5 ms pulse width for 240 seconds at 45 Volts. The temperature reached never exceeded 42˚C. After that, ESI procedures were performed. The patients received 2 mL of 1% lidocaine mixed with 4 mg dexamethasone. Withdraw the needle and apply a sterile bandage at the injection site. Check vital signs after the procedure.

Patients were evaluated 2 hours after the procedure and discharged with advices to avoid too much bending, lifting heavy weight, or walking long distances. Neurological check-ups of the lower extremities included motor tone, muscle strength, reflexes, and sensory examination.

## Outcome measurements

Patients' characteristics comprised sex, age, and baseline data upon admission. Baseline data were collected including VAS, ODI, and SLRT. Two experienced blinded doctors independently performed all assessments.

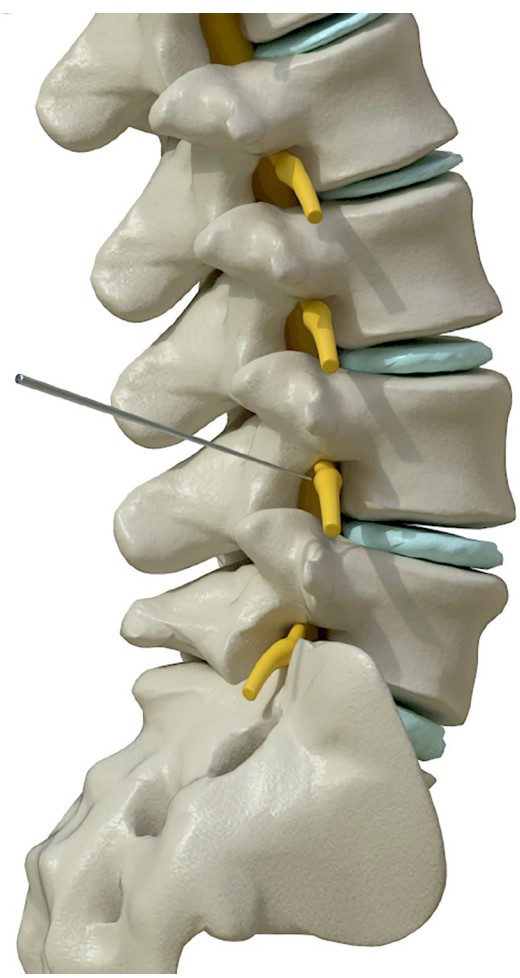

**Fig 2. 3D illustration of transforaminal PRF+ESI with lateral view.** Position the needle tip at the lower back of vertebrae foramen. Determine the depth of the needle tip by checking lateral view of the C-arm. Copyright kieutrinhnguyen. Published with permission.

Firstly, VAS was a method to evaluate the degree of pain. A 10 cm line as an indicator presented one end meaning no pain, while the other end meaning the most severe pain. The patient was asked to indicate the point on the line which could represent the patient's pain level.

VAS: before injection and after injection. To define as effective in pain relief, pain level post-op had to be smaller than pre-op by at least 2.5 points and had to decrease > 50% compared to pre-op.

Secondly, there were 10 items in the Oswestry disability index (ODI), including pain, individual function, and personal comprehensive function. The minimum score for each item is 0 (good state), whereas the highest score is 5 (poor state). ODI referred to the percentage of the sum of scores from all 10 items out of 50.

Thirdly, straight leg raising test scores:

- Pain up to 35 degrees is suggestive of prolapsed, herniated, or extruded intervertebral disc.

- 35–70 degrees is suggestive of disc prolapsed.

- Pain beyond 70 degrees is equivocal.

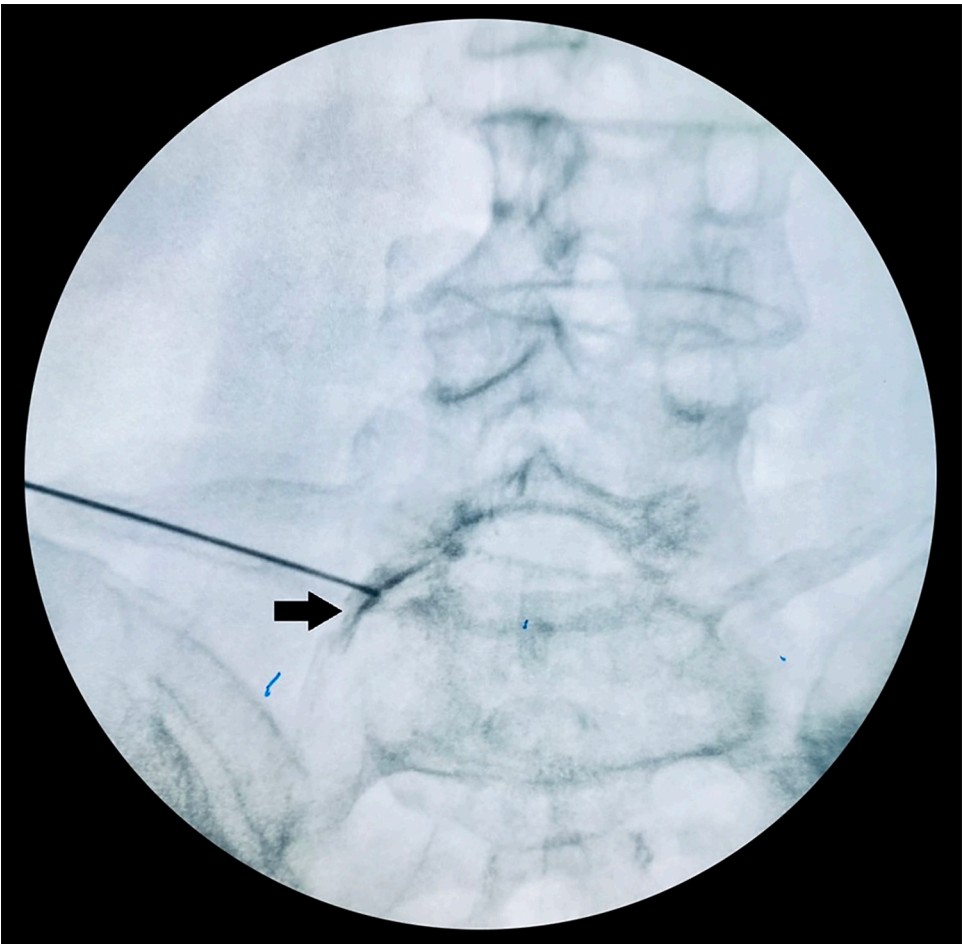

**Fig 3. Visualization of L5 nerve root by injecting of contrast agent into the anterior epidural region.**

Lastly, patients' satisfaction was subjectively recorded using a four-point verbal rating scale (0 = "very dissatisfied", 1 = "dissatisfied", 2 = "neutral", 3 = "satisfied", 4 = "very satisfied").

### Adverse events

PRF is a method with a high safety profile, without major risks, and with a high level of patient satisfaction. Minor complications have been described, such as pain during cannula placement, the subsequent increase in radicular pain or lower back pain, headaches, or postoperative discomfort. The root puncture has been described, which appears to be secondary to mechanical insult caused by the needle rather than by the application of RF itself. Other reported complications include local erythema and pain at the needle insertion site. Infectious complications, such as epidural abscesses, meningitis, and sepsis, have been reported in other categories of spinal interventional procedures, and therefore may potentially occur in dorsal root ganglion RF [7].

Adverse effects were carefully evaluated at each visit to detect pain flare-ups and newly developed neurologic deficits after the procedures.

## Statistical analysis

Mean, standard deviation, median, inter-quartile and frequency, percentage were used to describe the data. Chi-squared tests, Fisher's exact tests, Paired-Sample T-tests were used to evaluate the association between patients' characteristics and pain degree. All statistical tests were two-sided with a significance level of 0.05. Continuous data before operation with non-normal distribution was analyzed using the Mann–Whitney U-test. Data were analyzed using the SPSS 25.0 software.

## Ethics statement

The Ethics Committee of University of Medicine and Pharmacy at Ho Chi Minh City approved our research project (no. 3106/QĐ-BVĐHYD). All procedures were in accordance with the national ethical standards and with the 1964 Helsinki declaration and its later amendments. All participants read the participation information sheet and signed the written consent form.

## Results

### Patients' characteristics

Our study enrolled 76 patients for transforaminal PRF and ESI from November 2019 to November 2021 at University Medical Center, Ho Chi Minh City, Vietnam. Clinical data were collected before and after the procedure. Age ranged from 35 to 69 years old, with a mean age of 52.7 ± 15.2 years old. More than half were female patients (56%). At hospital admission, the median baseline of VAS was 7.0 (6.0–8.0). None of patients had SLRT>75 degree and mostly suffered severe disability (Table 1).

### Efficacy of transforaminal PRF and ESI with longitudinal data

VAS at all follow-up were significantly lower than the VAS baseline (p<0.001). For short-term at 1-month and 3-month follow-up, pain relief was achievable (had to decrease > 50% and at least 2.5 points). Fig 4 demonstrated that improvement. For longer term, low VAS scores

**Table 1. Patients' characteristics and baseline information upon admission.**

| Characteristics | | N = 76 |
|---|---|---|
| **Age** (year, mean ± SD) | | 52.7 ± 15.2 |
| $\geq$ 50 | | 47 (61.8%) |
| < 50 | | 29 (38.2%) |
| **Female** (%) | | 56 |
| **Time of radicular pain** (months, mean ± SD) | | 16.1 ± 9,5 |
| **Etiologies** | | |
| Intervertebral disc disorder | | 48 (63.2%) |
| Spinal stenosis | | 16 (21%) |
| Spondylolisthesis (grade I) | | 12 (15.8%) |
| **VAS** (median (1st quartile– 3rd quartile)) | | 7.0 (6.0–8.0) |
| **ODI** (%) | 21–40% | 0% |
| | 46–60% | 0% |
| | 61–80% | 25% |
| | 81–100% | 75% |
| **SLRT** (%) | >75˚ | 0% |
| | 35˚ - 75˚ | 71% |
| | < 35˚ | 29% |

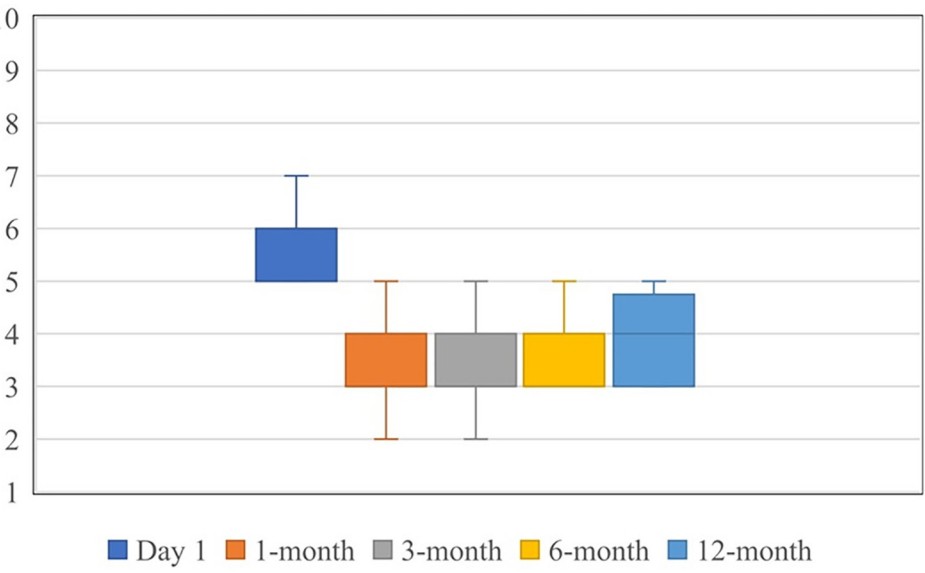

**Fig 4. Outcome measured by Visual Analogue Scores.**

remained dominant and lasting, however, Fig 4 also marked a slight increase of VAS score at 12-month follow-up.

Table 2 and Fig 5 showed an uptrend in ODI that implied a better quality of life through functional improvement. The positive results were almost spontaneous since day 1 and continuing through months of follow-up. However, of note, ODI chart similarly marked a fading effect at 12-month post-procedural.

The improvement of SLRT was found to be identical to VAS and ODI (Table 3 and Fig 6). Meanwhile, patients' satisfaction was ranged from 3 to 4 after treatment (Fig 7). There were no patients with a satisfaction score of zero.

No adverse events were observed after this procedure in the first 6 months, although a few patients had mild pain at the puncture site in the days following treatment. This discomfort was resolved spontaneously without further treatment in hospital. There were 3 patients lost to follow-up, 2 patients had undergone discectomy post-procedural 8 months, and no patients repeated the injection.

## Discussion

Nowadays, several methods are available for treating lumbosacral radicular pain, consisting of surgery and non-surgical treatment. Although the conservative/non-surgical treatment

**Table 2. Outcomes measured by Oswestry disability index.**

| Time<br>Level of disability | Baseline<br>(N = 76) | 1 week<br>(N = 76) | 1 month<br>(N = 76) | 3 months<br>(N = 76) | 6 months<br>(N = 76) | 12 months<br>(N = 76) |
|---|---|---|---|---|---|---|
| Minimal (0–20%) | 0.0% | 0.0% | 7.9% | 7.9% | 7.9% | 0.0% |
| Moderate (21–40%) | 0.0% | 25.0% | 55.2% | 59.2% | 63.1% | 55.3% |
| Severe (41–60%) | 25.0% | 55.2% | 36.9% | 32.9% | 29.0% | 44.7% |
| Cripple (61–80%) | 75.0% | 19.8% | 0.0% | 0.0% | 0.0% | 0.0% |
| Bed-bound (81–100%) | 0.0% | 0.0% | 0.0% | 0.0% | 0.0% | 0.0% |
| **p value** | | <0.001 | <0.001 | <0.001 | <0.001 | <0.001 |

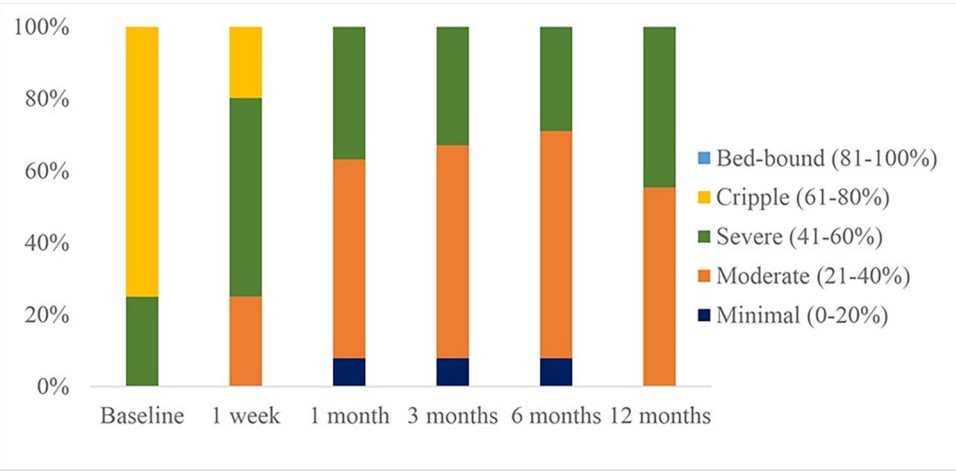

**Fig 5. Outcome measured by Oswestry disability index.**

provides many great advantages, it requires commitment and cooperation from patients to achieve long-term effect [8]. It also offers a poor curative effect and patients frequently suffers recurrent condition [9,10]. Therefore, minimally invasive interventional therapy has been widely used in treating chronic pain, and has gradually become the developmental direction for treating chronic lumbar neuropathic pain [10,11].

First of all, transforaminal PRF affects neuromodulation in the spinal nerve root, whose goals are to achieved a long-term analgesic effect and to prevent neural damage. The pulse of PRF stimulated the regulation of nerves and reduced the pain after nerve injury and mechanical oppression [12,13]. At dorsal root ganglion level, PRF enhances the expression of anti-inflammatory factors, such as GABAB-R1, Na/K-ATPase, and 5-HT3r; meanwhile, it decreases the expression of pro-inflammatory factors such as TNF-α and IL-6, and improving neuropathic pain [13]. Park and Chang [3] reported long-term depression of pain signaling from peripheral nerves to the central nervous system, inactivated microglia, decreased proinflammatory cytokines, increased endogenous opioid precursor mRNA, enhanced the descending pain inhibitory pathway. Therefore, the long-term analgesic effects of PRF are noteworthy.

Similarly, our findings recognized that ESI combined with transforaminal PRF displayed the characteristics of rapid onset and prolonged effect. Measurement of VAS showed spontaneous pain relief of transforaminal PRF+ESI intervention since day 1 post-op and was lasting for 12 months. Compare to body of literature, our findings were relatively more satisfying. For example, single PRF treatment for lumbar degenerative disc pain, the number of patients with >50% pain relief reached 56% one-year after [14]. Another method, intradiscal electrothermal therapy (IDET), patients were followed up for 6 months, and no significant difference was observed between the PRF and IDET groups [15]. Yang et al [5] reported that after a median

**Table 3. Outcome measured by Straight Leg Raising Test.**

| SLRT (°) | Time Baseline (N = 76) | 1 week (N = 76) | 1 month (N = 76) | 3 months (N = 76) | 6 months (N = 76) | 12 months (N = 76) |
|---|---|---|---|---|---|---|
| >75° | 0.0% | 32.8% | 76.3% | 82.6% | 80.2% | 69.7% |
| 35° - 75° | 71.0% | 48.7% | 23.7% | 17.4% | 19.8% | 30.3% |
| <35° | 29.0% | 18.5% | 0.0% | 0.0% | 0.0% | 0.0% |
| p value | | <0.001 | <0.001 | <0.001 | <0.001 | <0.001 |

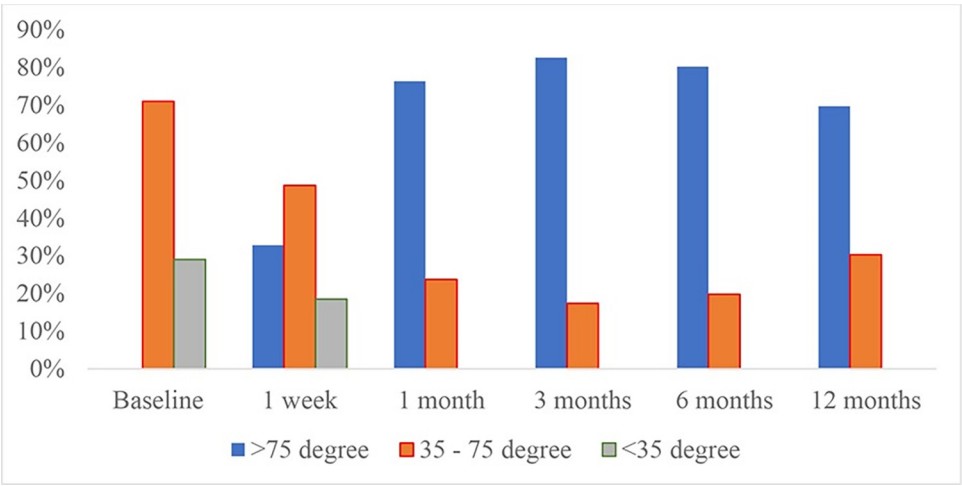

**Fig 6. Outcome measured by Straight Leg Raising Test.**

follow-up of 9.5 months, the VAS decreased from 6.5 ± 1.0 to 2.4 ± 1.9 at the 12-month follow-up. PRF combined with TESI is an effective approach to treat persistent lumbosacral radicular pain. The radical reasons behind our positive results were the resonance effect of transforaminal PRF+ESI, which satisfied both mechanical oppression and chemical physiopathology of chronic lumbar radiculopathy.

We also noted the trend toward better outcomes based on measurement of ODI and SLRT. Better SLRT and decreased ODI scores reflect functional improvement. The mean disability level observed pre-treatment was 75%, representing severe disability. Post-treatment, we reported moderate disability (55.3%) for 12 months. This result indicated that the procedure was effective in ameliorating spinal function. Besides, we also found that treated patients

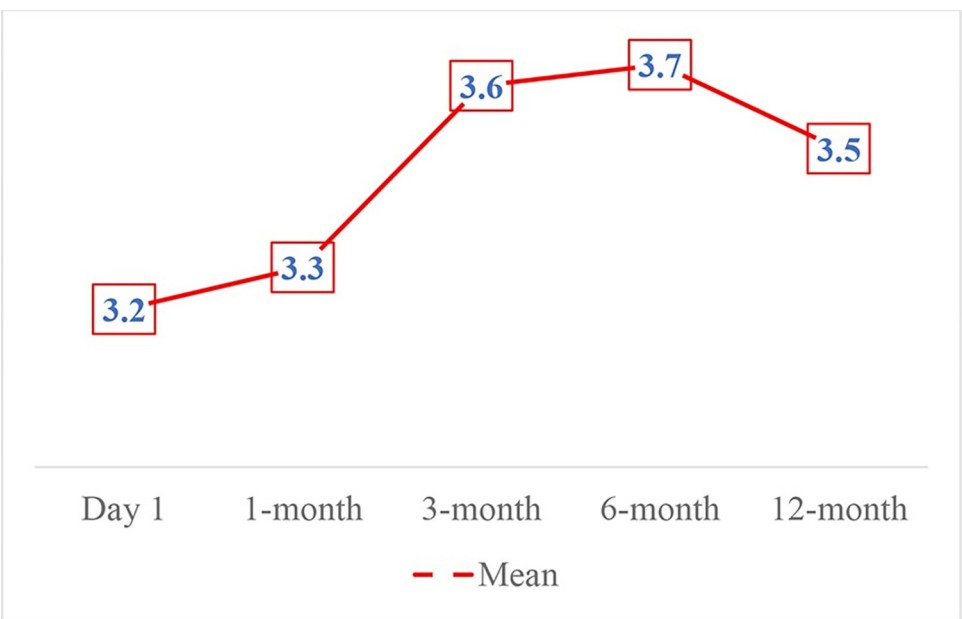

**Fig 7. Outcome measured by SAPS.**

reported higher satisfaction (mean: 3.5 after 12 months) without serious adverse event or persistent neurological deficits, and reduced pain medication consumption, leading to lower complication rate and lower cost. In a recent phase 3, multicenter, open-label, randomized controlled trial [16] recommends that in with sciatica secondary to herniated lumbar disc, with symptom duration of up to 12 months, transforaminal ESI should be considered as a first invasive treatment option. In an era of minimally invasive and economic conditions in Vietnam, this would potentially lumbar radicular pain management, resistant to medical treatments.

## Limitations

The limitations of our study were a single-center, and the lack of a control group. This study was designed as prospective and data-based research to investigate the effectiveness of transforaminal PRF and ESI in radicular pain. Long-term effects of PRP, the route of injections, the number of spinal segments for injection, and the interval between treatments are the issues that required a greater number of studies in the future.

## Conclusion

Transforaminal PRF combined ESI is a minimally invasive procedure offering modest, significant improvement out to 12 months follow-up average in the management of lumbosacral radicular pain. It is a promising technique and it has shown good results in providing intermediate to long-term relief of pain.

## Supporting information

**S1 Fig. Anteroposterior view of degenerative lumbar spine on C-arm image.** *Red arrow*: Hypertrophy of L4L5 facet join. RF canula was below the facet joint to enter inter-peduncle space. *Yellow arrow*: Inter-vertebrae foramen of L5S1 segment.
(TIF)

**S2 Fig. Lateral view of degenerative lumbar spine on C-arm image.** *Red arrow*: Inter-vertebrae foramen of L5S1 segment. *Yellow star*: Pedicle of L5 vertebrae. *Black arrow*: RF canula tip was check under C-arm.
(TIF)

**S1 Graphical abstract.**
(TIF)

**S1 Data.**
(XLSX)

## Acknowledgments

We would like to acknowledge kieutrinhnguyen (kieutrinh.artstation.com) for her assistance in creating illustrations: Fig 1 (https://flic.kr/p/2nDgA3o), Fig 2 (https://flic.kr/p/2nDhPpT). All rights reserved. Published with permission.

Illustrations based on the 3D model: "Spine (collection of thunthu)" (https://skfb.ly/CyHw) by thunthu is licensed under Creative Commons Attribution (http://creativecommons.org/licenses/by/4.0/).

## Author Contributions

**Conceptualization:** Viet-Thang Le, Phuoc Trong Do.

**Data curation:** Viet-Thang Le, Phuoc Trong Do.

**Formal analysis:** Viet-Thang Le, Phuoc Trong Do, Vu Duc Nguyen, Luan Trung Nguyen Dao.

**Methodology:** Phuoc Trong Do, Vu Duc Nguyen.

**Software:** Luan Trung Nguyen Dao.

**Visualization:** Luan Trung Nguyen Dao.

**Writing – original draft:** Viet-Thang Le, Phuoc Trong Do.

**Writing – review & editing:** Phuoc Trong Do, Vu Duc Nguyen, Luan Trung Nguyen Dao.

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
