## [Decision Letter · Decision Letter 0]

16 Aug 2023

PONE-D-22-31448Transforaminal Pulsed radiofrequency and Epidural steroid injection on chronic lumbar radiculopathy: A case series from a tertiary care hospital in VietnamPLOS ONE

Dear Dr. Nguyen Dao,

Thank you for submitting your manuscript to PLOS ONE. After careful consideration, we feel that it has merit but does not fully meet PLOS ONE’s publication criteria as it currently stands. Therefore, we invite you to submit a revised version of the manuscript that addresses the points raised during the review process.

We look forward to receiving your revised manuscript.

Kind regards,

Ipek Saadet Edipoglu

Academic Editor

PLOS ONE

Journal Requirements:

Reviewers' comments:

Reviewer's Responses to Questions

**Comments to the Author**

1. Is the manuscript technically sound, and do the data support the conclusions?

Reviewer #1: Yes

Reviewer #2: Partly

2. Has the statistical analysis been performed appropriately and rigorously? 

Reviewer #1: Yes

Reviewer #2: I Don't Know

3. Have the authors made all data underlying the findings in their manuscript fully available?

Reviewer #1: Yes

Reviewer #2: Yes

4. Is the manuscript presented in an intelligible fashion and written in standard English?

Reviewer #1: Yes

Reviewer #2: Yes

5. Review Comments to the Author

Reviewer #1: The authors investigated the effects of transforaminal Pulsed radiofrequency and epidural steroid injection on chronic lumbar radiculopathy. It was a well written study. I thank the authors for their manuscript presentation. In my opinion the study seems acceptable.

Reviewer #2: It is written as "neuropathic pain" in the abstract (conclusion) part. However, there is no neuropathic pain assessment scale in the material method section. For this reason, it would be more appropriate to say "pain" instead of neuropathic pain.

The phrase "A case series" in the title partially contradicts the statement "A prospective observational study" in the material method section. The title or abstract should state that the study is a prospective observational study.

Reference should be added for the "Study procedure". Has this procedure been applied in accordance with the literature?

Spondylolisthesis is included in the exclusion criteria, but in the result section, it is written that 12 (15.8%) patients have Spondylolisthesis. This is a contradiction.

Thanks

6. PLOS authors have the option to publish the peer review history of their article (what does this mean?). If published, this will include your full peer review and any attached files.

Reviewer #1: No

Reviewer #2: No

---

## [Author Response · Author response to Decision Letter 0]

20 Aug 2023

Thank you for providing thoughtful and valuable feedback on the manuscript. We have responded to all reviewer’ comments below and have changed the text accordingly. We believe this has strengthened the manuscript.

REVIEWER 1

1. It is written as "neuropathic pain" in the abstract (conclusion) part. However, there is no neuropathic pain assessment scale in the material method section. For this reason, it would be more appropriate to say "pain" instead of neuropathic pain.

Response: Thank you for your valuable comments. We have adjusted the term:

“Transforaminal PRF combined with ESI in management of lumbosacral radiculopathy should be a good method of choice for its effectiveness and safety in management of pain.” (page 2, line: 19-21).

2. The phrase "A case series" in the title partially contradicts the statement "A prospective observational study" in the material method section. The title or abstract should state that the study is a prospective observational study.

Response: We have made changes in title to match the Methods of our study:

“Transforaminal Pulsed radiofrequency and Epidural steroid injection on chronic lumbar radiculopathy: A prospective observational study from a tertiary care hospital in Vietnam.” (Title page)

3. Reference should be added for the "Study procedure". Has this procedure been applied in accordance with the literature?

Response: We have added more details: 

 “The stepwise procedure was conducted, according to the practice guidelines of the International Spine Intervention Society[6]” (page: 5, line: 92-93)

6. International Spinal Intervention Society. Lumbar transforaminal injections. In: Bogduk N, ed. Practice Guidelines for Spinal Diagnostic and Treatment Procedures. San Francisco, CA: International Spinal Inter-vention Society; 2004:163–87.

4. Spondylolisthesis is included in the exclusion criteria, but in the result section, it is written that 12 (15.8%) patients have Spondylolisthesis. This is a contradiction.

Response: We have added missing information:

“Spinal tumors, tuberculosis, spondylolisthesis (grade ≥ II), and fracture spine.” (page 4, line: 71)

Thank you very much for your comments and suggestions!

---

## [Editor Report · Decision Letter 1]

12 Sep 2023

Transforaminal Pulsed radiofrequency and Epidural steroid injection on chronic lumbar radiculopathy: A prospective observational study from a tertiary care hospital in Vietnam

PONE-D-22-31448R1

Dear Dr. Nguyen Dao,

We’re pleased to inform you that your manuscript has been judged scientifically suitable for publication and will be formally accepted for publication once it meets all outstanding technical requirements.

Kind regards,

Ipek Saadet Edipoglu

Academic Editor

PLOS ONE
---

## [Editor Report · Acceptance letter]

20 Sep 2023

PONE-D-22-31448R1 

Transforaminal Pulsed radiofrequency and Epidural steroid injection on chronic lumbar radiculopathy: A prospective observational study from a tertiary care hospital in Vietnam 

Dear Dr. Nguyen Dao:

I'm pleased to inform you that your manuscript has been deemed suitable for publication in PLOS ONE. Congratulations! Your manuscript is now with our production department. 

Kind regards, 

on behalf of

Dr. Ipek Saadet Edipoglu 

Academic Editor

PLOS ONE